

# Are the effects of vegetation and soil changes as important as climate change impacts on hydrological processes?

Kabir Rasouli[1, 2], John W. Pomeroy[2], and Paul H. Whitfield[2, 3]

[1]Department of Geoscience, University of Calgary, Calgary, AB, T2N 1N4, Canada

5  [2]Centre for Hydrology, University of Saskatchewan, Saskatoon, SK & Canmore, AB, T1W 3G1, Canada

[3]Environment and Climate Change Canada, Vancouver, BC, Canada

*Correspondence to*: Kabir Rasouli (kabir.rasouli@ucalgary.ca)

**Abstract.** Hydrological processes are widely understood to be sensitive to changes in climate, but the effects of changes in vegetation and soils have seldom been considered. The response of mountain hydrology to future climate and vegetation/soil changes is modelled in three snowmelt dominated mountain basins in the North American Cordillera. A Cold Regions Hydrological Model developed for each basin was driven with perturbed observed meteorological time series. Monthly perturbations were developed from differences in eleven regional climate model outputs between the present and future scenarios. Future climate change in these basins results in decreased modelled peak snow water equivalent (SWE) but increased evapotranspiration in all basins. All three watersheds became more rainfall-dominated. In Wolf Creek in the Yukon Territory, an insignificant increasing effect of vegetation change on peak SWE was found to be important enough to offset the significant climate change effect on alpine snow. In Marmot Creek in the Canadian Rockies, a combined effect of soil and climate changes on increasing annual runoff becomes significant while their individual effects are not statistically significant. In the relatively warmer Reynolds Mountain East catchment in Idaho, USA, only vegetation change decreases annual runoff volume and changes in soil, climate, or combination of them do not affect runoff. At high elevations in Wolf and Marmot Creeks, modelled vegetation/soil changes moderated the impact of climate change on peak SWE, the timing of peak SWE, evapotranspiration, and annual runoff volume. At medium elevations, these changes intensified the impact of climate change, decreasing peak SWE, and sublimation. The modelled hydrological impacts of changes in climate, vegetation, and soil in mountain environments are similar in magnitude but not consistently in the direction in all biomes; in some combinations, this results in enhanced impacts at lower elevations and latitudes and offsetting effects at higher elevations and latitudes.

## 1 Introduction

Under warmer and drier conditions, vegetation is expected to change in temperate mountain climates such as French Alps, which will result in evapotranspiration increases (Beniston, 2003). In subarctic mountains, warming degrades permafrost and leads shrub tundra expansion (Tape et al., 2006; Hallinger et al., 2010). The larger shrub extent traps more windblown snow, increases snowmelt volumes, lowers spring albedo, and alters melt rates (Pomeroy et al., 2006; Krogh and Pomeroy, 2018). The changes in
ice/snowcover, runoff, soils, vegetation, and other components of an ecosystem lead to important local and global feedbacks in the ecohydrology and energy budgets (Osterkamp et al., 2009; Rawlins et al., 2009). During vegetation change soil properties vary in phases from the initial colonization of the bare surface to the establishment of a forest (Crocker and Major, 1955) and soil



development may not be as quick as vegetation change (Innes, 1991). Potential changes in soil, especially changes in organic matter content under vegetation changes can be important and might affect infiltration, groundwater recharge, annual runoff

volume, and floods (DeBano, 1991). Therefore, soil changes also need to be considered in hydrological modelling along with the climate and vegetation changes. Potential changes in soil properties, including porosity and depth, as a result of vegetation change, were also examined in this study. Three scenarios are applied to identify whether changes in vegetation, soil, or combination of them are important in affecting hydrological processes. These scenarios are: (1) only vegetation changes and soils remain unchanged; (2) only soil changes and vegetation remains unchanged and (3) both vegetation and soils change. In mountains, the

major drivers of vegetation change are climate, mountain pine beetle (in western Canada), logging, and wildfires. One example of how the interaction between climate and vegetation can change ecosystems is the expansion of shrubs in northern latitudes (Martin et al., 2017; Myers-Smith and Hik, 2018). Bjorkman et al. (2018) reported that the height of the tundra community has increased with warming over the 1989–2015 period in more than 100 tundra sites. Bosch and Hewlett (1982) reviewed the impacts of deforestation and afforestation on water yield in forested landscapes and concluded that, on average, water yield increases in

coniferous forests (e.g., pine) by 40 mm, in deciduous hardwood forests by 25 mm, and in shrubs by 10 mm per each 10 % reduction in cover; the maximum increases occur in the first five years following forest cover removal. The growth rates of trees have increased (Innes, 1991), the forest composition (e.g., in the Pacific Northwest) has changed (Dale and Franklin, 1989), and tree-line (Hansell et al., 1971) has moved vertically and Northwards within the last century. Neilson and Marks (1994) found a consistent regional pattern between vegetation and annual runoff changes. Vegetation changes can alter soil properties.

In cold regions in general and mountains in particular, the amount and timing of snowmelt affect soil moisture, nutrient transport, soil and leaf temperature, surface microclimate, and growing season (Billings and Bliss, 1959; Walker et al., 1993; Stanton et al., 1994). Vegetation response to warming varies from one climate to another (Stow et al., 2004). In northern latitudes, where air temperature is low, growing season is short, cloud cover is persistent, and solar angle is small, the vegetation composition responds quickly to changes in climate and nutrient availability; with warming, rapid changes in thawing and freezing processes (Zhang et

al., 2008; Walvoord and Kurylyk, 2016), snowmelt rates, and soil moisture (Bales et al., 2011) are expected. In more moderate climates the growing season is longer. Many mountain plants begin growth at near-freezing temperatures when snowpacks start to melt (Billings and Bliss, 1959), and snow depth and snowmelt rates affect vegetation composition (Billings and Bliss, 1959; Stanton et al., 1994). In a warmer climate, the abundance of cold-adapted species decreases and warmth-demanding vegetation expands into higher elevations (Lamprecht et al., 2018) and plant communities shift to more northern latitudes (Alberta Natural

Regions Committee, 2006; Schneider et al., 2009; Mann et al., 2012; Schneider, 2013; Myers-Smith and Hik, 2018).

Vegetation and soil changes under climatic changes in cold regions alter soil moisture, streamflow, snowcover, and permafrost (DeFries and Eshleman, 2004; Osterkamp et al., 2009). Conversion of forest to pasture can increase the soil bulk density and decrease the soil porosity, both of which alter infiltration, percolation, aeration, and erodibility (Reiners et al., 1994). Change in active layer thickness, as a result of the warming climate, allows more subsurface water storage, higher nutrient transport, and a

deeper root zone leading to the expansion of shrubs (Sturm et al., 2005).

The climate perturbation method, also known as the delta change factor method, has been widely used in the climate change impact studies (e.g., Rasouli et al., 2014, 2015). This method avoids the computational cost of the dynamical downscaling and maintains consistency in relationships of the atmospheric fields, which may be distorted in statistical methods if the interaction of the variables is not considered (Hijmans et al., 2005; Gutmann et al., 2016). Sub-monthly and daily values obtained from regional



climate models (RCMs) are highly uncertain and contain a large amount of noise, especially in mountainous regions. The number of dry days simulated by RCMs for the present period is also higher than those observed in the basins studied here (Rasouli, 2017). Unlike using the RCM outputs directly, the change factor based climate perturbation approach produces spatial and seasonal precipitation based on observations and changes based on simulated differences (Hay et al., 2000; Kay et al., 2009; Sunyer et al., 2012). This represents weather with reasonable accuracy and also represents observed extreme dry periods and observed extreme

storms. Limitations of applying change factors of the monthly climatological values to perturbed climate are that any future changes in weather patterns, extremes, sequences of wet or dry spans, and droughts or floods are not adequately represented. This is similar to the assumption of the stationary relationships between large-scale circulations and locally observed data in statistical downscaling. Changes in synoptic dynamics of the atmosphere cannot be captured by the climate perturbation method, nor can RCM capture local-scale processes in mountainous regions (Rasouli, 2017).

There have been many studies on the impact of climate change on hydrology and some on mountain hydrology (e.g. Wolf Creek Yukon Territory: Pomeroy et al. (1999, 2003, 2006); Rasouli et al. (2014); Williams et al. (2015); Marmot Creek, Alberta: Pomeroy et al. (1999, 2012, 2015, 2016); Fang et al. (2013); and Reynolds Mountain East, Idaho: Link et al. (2004); Flerchinger et al. (2012)). Modelling experiments (Rasouli et al., 2019) show that future climates are warmer and wetter, especially in the northern latitudes and have complex effects in snow-dominated watersheds. Warmer and wetter future conditions are expected to drive

vegetation changes, and hydrological changes but such changes have not been studied thoroughly. There is limited knowledge about the impacts of climate and vegetation changes on hydrological processes in cold or even mild climates (Brown et al., 2005). Rasouli et al. (2019) found that reduced snowfall amounts are offset by reduced losses due to snow sublimation and increased rainfall amounts are offset by increased evapotranspiration, both leading to no change in annual runoff in Marmot Creek and Reynolds Mountain East basins despite increased precipitation and rainfall amounts.

Plausible vegetation changes, adapted from Alberta Natural Regions Committee (2006); Schneider et al. (2009); and Myers-Smith and Hik (2018), were applied to each basin. In Wolf Creek, the changes were: (1) an upward movement of the treeline and shrub tundra expansion, (2) only shrub tundra expansion, and (3) only an upward movement of the treeline. In Marmot Creek, the changes were (1) an upward movement of the treeline, afforestation of the harvested forest, and deforestation of the lower elevations, (2) an upward movement of the treeline and afforestation of the harvested forest, and (3) afforestation of the harvested

forest and deforestation of the lower elevations. In Reynolds Mountain East, the changes were: (1) deforestation of all trees (aspen, fir, willow) and mountain sage expansion, (2) deforestation of fir trees and mountain sage expansion, and (3) deforestation of aspen and fir trees and mountain sage expansion. These changes are conceptualized in Fig. 1.





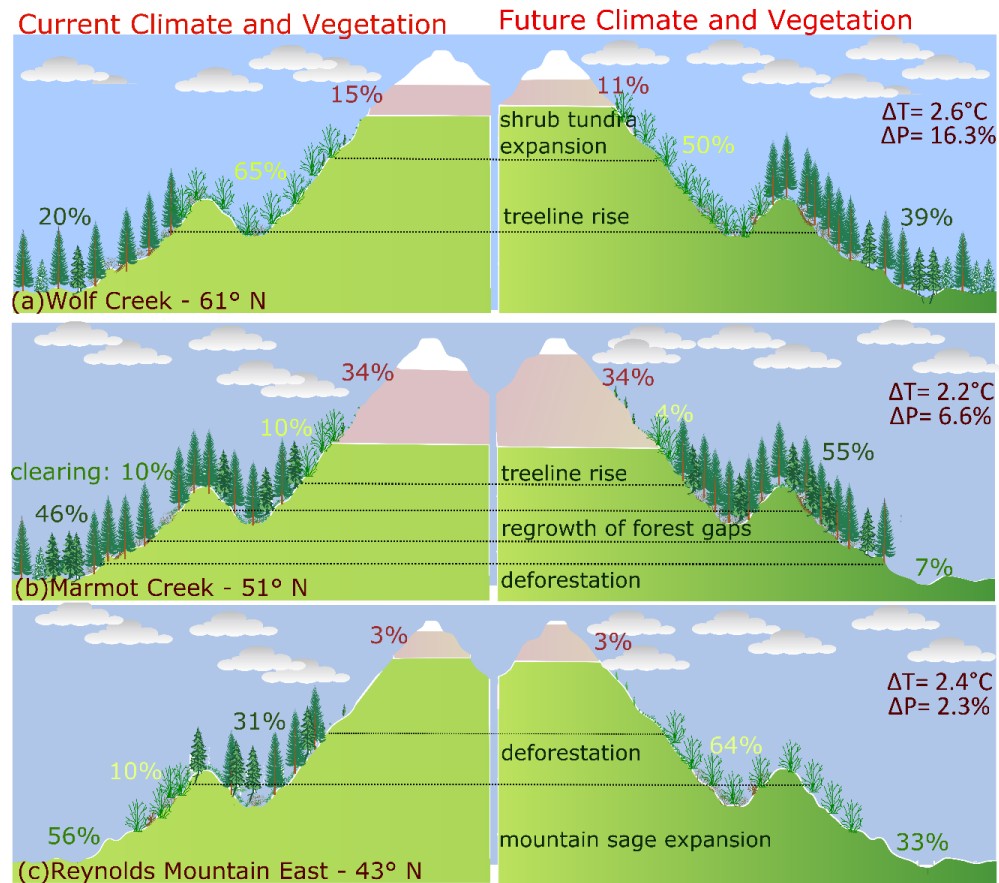

**Figure 1.** Schematic illustration of the vegetation cover under current and monthly perturbed climates in the Wolf Creek Research Basin (WCRB), Marmot Creek Research Basin (MCRB), and Reynolds Mountain East (RME) catchment along a north–south transect of the North American Cordillera. The numbers show the areal percentage of alpine, forest, shrub tundra, grassland, and forest clearing biomes. ΔT and ΔP, which are projected by eleven regional climate models, denote annual warming and precipitation increases, respectively (Rasouli et al., 2019).

Interactions between climate, vegetation, and soils are complex (Rodriguez-Iturbe, 2000) and the time lag between vegetation response to climate changes and soil response to vegetation changes is unclear (Innes, 1991). Under a warmer climate with a longer snow-free season and with increased precipitation in northern latitudes, vegetation is expected to increase in valley bottoms where adequate soil moisture and nutrients are available. Assuming there will be no change in vegetation when modelling mountain hydrology in the present and future climates if changes in vegetation affect the hydrology could result in an incorrect assessment. Modelling climate change effects on hydrology with and without vegetation change can help to understand separate and combined effects of climate and vegetation changes in mountainous watersheds. Simulations of the future hydrological conditions in the mountains are challenging because of the large biases between climate model outputs and locally observed





hydroclimatic conditions and the seasonal nature of snow accumulation and depletion (Fowler et al., 2007; Bennett et al., 2012). To overcome this, we use the delta method to perturb observations using the monthly difference between RCM modelled present

and future.

## 2 Methods

### 2.1 Study sites and data sources

Three mountain basins across the North American Cordillera (NAC) are examined: a sub-Arctic basin (Wolf Creek Research Basin, WCRB ~61°N, Yukon Territory), a headwater catchment in the Canadian Rockies (Marmot Creek Research Basin, MCRB

~51°N, Alberta), and a small catchment with cool montane climate (Reynolds Mountain East catchment, RME ~43°N, Idaho) (Fig. 2).  WCRB has the shortest distance to the Pacific Ocean (Fig. 2), lowest average elevation, coldest climate, and lowest annual precipitation amongst the three basins.  MCRB has the highest elevation and highest annual precipitation and wind speed.  RME has the smallest drainage area, highest average elevation, and lowest wind speed (Table 1).

Jack pine, spruce, and aspen forests are dominant vegetation types at low elevations in WCRB (Francis et al., 1998) and 65% of

the basin area above the forest biome is covered with birch and willow shrub tundra with heights ranging between 30 cm to 2 m. Alpine tundra with short moss, grass, and bare rock covers high elevations in WCRB.  Engelmann spruce and subalpine fir cover high elevations and lodgepole pine stands cover low elevations in MCRB (Kirby and Ogilvie, 1969).  Areas adjacent to the treeline in MCRB are covered with shrubs and alpine larch.  Spatial variability of vegetation is large within RME (Seyfried et al., 2009; Winstral and Marks, 2014) and grass, mountain sagebrush, riparian willow, aspen, and coniferous trees are dominant vegetation

types in this basin. Almost 43 % of WCRB is covered by continuous and discontinuous permafrost (Lewkowicz and Ednie 2004). Soils do not freeze in RME and freeze seasonally in MCRB.

Precipitation was measured by tipping bucket, unshielded "BC style standpipe", and shielded Nipher gauges in WCRB, by an Alter-shielded Geonor gauge in MCRB, and by shielded and unshielded gauges in RME. Precipitation data were adjusted using a wind undercatch correction equation (Goodison et al., 1998) based on wind speeds measured at the same height that precipitation

was measured.  In addition to precipitation measurements, air temperature, humidity, wind, shortwave radiation, and streamflow data have been collected at hourly time steps for each basin.



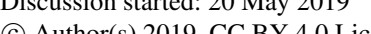

**Figure 2.** Three headwater basins across the Western North American Cordillera: **(a)** Wolf Creek Research Basin, Yukon Territory; **(b)** Marmot Creek Research Basin (MCRB); and **(c)** Reynolds Mountain East (RME) catchment within Reynolds Creek Experimental Watershed, Idaho, USA (more details are provided in Rasouli et al., 2019).

## 2.2 Modelling strategy

The effects of vegetation and soil changes on snow regimes and hydrological variables were evaluated under conditions in which: (1) climatic conditions change assuming no changes in future vegetation will occur, (2) no changes in future climate but



vegetation/soil changes will occur, and (3) changes in future climate will be accompanied by vegetation/soil changes. Eight scenarios for different combinations of changes in climate, soil, and vegetation are compared under present and future climate forcings for the three watersheds to allow differentiation between the effect of changes in climate, vegetation, and soils.

A distributed hydrological model for each basin was developed on the CRHM platform based on basin physiography, elevation bands, ranges of slope and aspect, and vegetation types in biomes. The models were developed with observed data and then used to investigate the response of hydrological fluxes to climate, vegetation, and soil changes using perturbed observations, described

below. These CHRM models consist of physically based subroutines that describe the major hydrological mechanisms in cold regions; this approach has been used extensively (e.g., Pomeroy et al., 1999; MacDonald et al., 2009). The model is based upon hydrological response units (HRUs) that are spatially segregated based on model parametrization including vegetation type, topographic slope and aspect, soil depth, layers, and the variability of basin attributes. The CRHM models are run on all HRUs at hourly time steps (Table 1). HRUs have different sizes. Details on model parametrization and performance are available in Rasouli

et al. (2014, 2015) and Rasouli (2017).

Hydrological model parameters under vegetation and soil changes are based on parameters that represent vegetation and soil characteristics under current climate. For example, an HRU in the model for the present is covered by grass, and it is expected to be forest in the future climate, the vegetation parameters for that HRU will be changes to those of an HRU that is presently covered by forest. HRU parameters are transferred to represent three different changes (i) only transferring vegetation parameters, (ii) only

transferring soil parameters, and (iii) transferring both vegetation and soil parameters. This allows separation of the effect above ground changes from changes in the characteristics in soils. Changes in the organic layer of soils following transient vegetation changes can alter the soil characteristics including soil macro-pores and hence, alter snowmelt/rainfall infiltration, thawing/freezing processes, recharge into groundwater, and runoff mechanisms. The soil porosity in different soil layers and soil depth were two model parameters that were changed under soil change scenarios.


### 2.3 Perturbed observations

Monthly perturbed climates are constructed from the dynamical downscaling method applying delta changes in monthly climatology to baseline hourly observations; see Rasouli et al. (2019) for details. The monthly perturbation was constructed based on eleven climate models from the North American Regional Climate Change Assessment Program (NARCCAP, Mearns et al.,

2007), which are driven by outputs from multiple general circulation models (GCMs). Using observed data modified by the monthly delta gives an estimate of the potential climate change impacts on these driving forces consistent with the large-scale atmospheric circulations. The deltas were estimated by the difference between the current (1970–2000) monthly 30-year climatology from future (2041–2070) monthly 30-year climatology.

### 2.4 Significance testing

Significant changes and differences in water balance components, snow characteristics, and their timing (initiation date, peak SWE date, snow-free date, and duration of snowcover season) between simulations under the present period (CVS_) and simulations



under different scenarios were assessed with the nonparametric Mann–Whitney U-test (Wilcoxon, 1945; Mann and Whitney, 1947). The differences between simulated distributions in the modelled present period for $n$ years ($x_{1:11 \times n}^{c}$, $11 \times n$ values) and the simulated distributions in the modelled future periods, obtained for eleven RCMs ($x_{1:11 \times n}^{f}$, $11 \times n$ values) were determined.

Assessment of the changes in the hourly SWE distribution due to vegetation changes was done with the nonparametric two-sample Kolmogorov-Smirnov test (Massey, 1951) was used. This test evaluates the difference between the cumulative density functions of the hourly SWE in the present period and a climate or vegetation alternative. Using an analysis of variance on the annual differences between the modelled scenarios and the modelled present (CSV_) and the honestly significant difference test (Tukey, 1991) for each basin, differences in snow and runoff under the following four groups and eight scenarios were determined:

Group (1):

- present climate, present vegetation, and present soil (CVS_ )

Group (2):

- present climate, future vegetation, present soil (CS_V)
- present climate, present vegetation, future soil (CV_S)
- present climate, future vegetation, future soil (C_VS)

Group (3):

- future climate, present vegetation, present soil (VS_C)


Group (4):

- future climate, future vegetation, present soil (S_VC)
- future climate, present vegetation, future soil (V_SC)
- future climate, future vegetation, future soil (_CVS)


The interannual variability, indicated by the 95% confidence intervals, was calculated for snow and streamflow based on standard deviation and number of years (18 for WCRB, 8 for MCRB, 25 for RME), assuming a standard normal distribution.

**3 Results**

**3.1 Synergic effects of climate, vegetation, and soil changes on snow and runoff regimes**

Changes in peak SWE and annual runoff volume due to vegetation, soil, and their interaction in the present climate were compared with the modelled present (present climate – present vegetation – present soil simulations (CVS_)) to detect the effect of individual or combined changes and find whether combinations of changes can offset each other's effects. Similarly, changes in peak SWE and runoff due to changes in vegetation, soil, and their interaction in the future climate were compared with future-climate scenarios as well as present-climate scenarios. In total, four scenarios under present climate and four scenarios under future climate were





studied and statistically different scenarios, based on the Tukey's honestly significant difference test, were distinguished from the CVS_ scenario (modelled present), and all scenarios were classified into multiple groups for each variable (Fig. 3, 4 & 5). The star signs in Fig. 3 , 4 & 5 differentiate significantly different scenarios from the CVS_ scenario.

The snow regime in WCRB shows a significant response to climate change in the alpine biome (Fig. 3a), significant changes to both vegetation and climate changes in the shrub tundra biome (Fig. 3b), and no significant response to vegetation and climate changes in the forest biome (Fig. 3c). In the alpine biome within WCRB, snow is significantly decreased with climate change. Therefore, snow regime under the future climate scenario (VS_C) is statistically different from the CVS_ scenario (Fig. 3a). The vegetation change effect on peak SWE in this biome, however, offsets the climate effect as two CVS_ and S_VC scenarios are not statistically different, and both are classified into group "b" (Fig. 3a). This shows that vegetation change effect on increasing peak SWE (Fig. 3a) is not statistically different from the present simulations, but it is important enough to offset the climate change effect on alpine snow in WCRB. In contrast to the forest biome in WCRB, which is not affected by any changes (Fig. 3c), the decrease in peak SWE in the shrub tundra biome by climate change is intensified under both climate and vegetation changes (Fig. 3b). Soil changes do not affect peak SWE in WCRB. The annual runoff regime in WCRB shows significant responses to individual vegetation, soil, and climate changes (Fig. 3d). The individual soil and vegetation changes decrease annual runoff while climate change increases annual runoff volume in WCRB (Fig. 3d). Two S_VC and _CVS scenarios are classified into the same group as the present simulations ( CVS_ ), group "c", suggesting that the effects of the changes on runoff in WCRB are counteracting and the increasing effect of climate change on runoff is offset by decreasing effects of vegetation and soil changes. The CS_V and C_VS scenarios and the VS_C and V_SC scenarios are classified into the "b" and "d" groups in Fig. 3d, respectively, suggesting that vegetation and climate changes become dominant in WCRB when concurring with soil changes.

The snow regime in MCRB show no significant response to vegetation and climate changes in the high elevation alpine biome (Fig. 4a), a significant response to climate change in the low elevation forest biome (Fig. 4b), and significant responses to individual vegetation and climate changes and their interactions in the forest clearing (Fig. 4c) and treeline biomes (Fig. 4d). The vegetation change decreases the peak SWE significantly in the forest clearing (Fig. 4c) and treeline biomes (Fig. 4d), leading to a decrease in annual runoff in MCRB (Fig. 4e). Soil changes do not affect peak SWE in MCRB. The annual runoff regime in MCRB decreases with vegetation change and increases with a combined soil– climate change (Fig. 4e). The response of annual runoff to individual soil and climate change effects is not significantly different from the present simulations (all three CVS_, CV_S, and VS_C are classified into the "b" group), but it is different under a combined effect of climate and soil changes (V_SC is classified into group "c" in Fig. 4e). Therefore, the soil-climate interaction is more important in changing annual runoff in MCRB than the individual effects of soil and climate.

The snow regime in RME show a significant response to climate change in the alpine biome (Fig. 5a), and significant changes to both vegetation and climate changes in the forest biome (Fig. 5b), blowing wind sheltered zones (Fig. 5c), and blowing snow sink zones (Fig. 5d). The peak SWE in all of the biomes in RME show significant changes under climate, vegetation, and combination of these two (Fig. 5), except for the alpine biome, which shows only significant response to climate change (Fig. 5a). Similar to the other two basins, soil changes do not affect peak SWE in RME. Soil and climate changes do not affect the annual runoff while vegetation change significantly decreases the annual runoff in RME (Fig. 5e). Even though the individual effect of soil change on runoff is not statistically significant, its effect, however, can enhance the decreasing effect of the vegetation change in this basin (Fig. 5e).




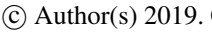

**Figure 3.** Differences in peak snow water equivalent (SWE) and annual runoff volume in the Wolf Creek Research Basin relative to present climate, present vegetation, and present soil (CVS_). Indices before and after the "_" sign show present conditions with no change and future conditions with changes, respectively. Stars denote which scenarios are significantly different from the CVS_ scenario using the Tukey's honestly significant difference test. Letters are used to compare and group similar scenarios. The shaded scenarios on the right-hand side of the plot demonstrate vegetation and soil changes under modelled future climate while the unshaded scenarios on the left-hand side of the plot demonstrate changes under modelled present climate.






**Figure 4.** Differences in peak snow water equivalent (SWE) and annual runoff volume in the Marmot Creek Research Basin relative to present climate, present vegetation, and present soil (CVS_). Indices before and after the "_" sign show present conditions with no change and future conditions with changes, respectively. Stars denote which scenarios are significantly different from the CVS_ scenario using the Tukey's honestly significant difference test. Letters are used to compare and group similar scenarios. The shaded scenarios on the right-hand side of the plot demonstrate vegetation and soil changes under modelled future climate while the unshaded scenarios on the left-hand side of the plot demonstrate changes under modelled present climate.







**Figure 5.** Differences in peak snow water equivalent (SWE) and annual runoff volume in the Reynolds Mountain East relative to present climate, present vegetation, and present soil (CVS_). Indices before and after the "_" sign show present conditions with no change and future conditions with changes, respectively. Stars denote which scenarios are significantly different from the CVS_ scenario using the Tukey's honestly significant difference test. Letters are used to compare and group similar scenarios. The shaded scenarios on the right-hand side of the plot demonstrate vegetation and soil changes under modelled future climate while the unshaded scenarios on the left-hand side of the plot demonstrate changes under modelled present climate.






The significant change under vegetation change scenario (CS_V scenarios in Figures 3-5) shows that vegetation change in all three basins has an important effect on snow and runoff regimes, except for the snow regimes in the alpine and forest biomes in MCRB.

Different plausible mechanisms of vegetation change in each basin are examined to better diagnose the effects of the changes. Figure 6 shows the differences between the annual cycle of hourly modelled SWE for the present simulations and the three vegetation scenarios for each basin. All SWE values are modelled under current climate with no changes in soil. Snow regimes are mainly sensitive to the upslope treeline movement and to a lesser extent to shrub tundra expansion in WCRB (Fig. 6a), to treeline movement and infill of the harvested forest in MCRB (Fig. 6b), and willow tree deforestation in the riparian zone and

mountain sage expansion in RME (Fig. 6c). Deforestation of fir and aspen biomes in RME prolongs the length of the snowcover season and increases the basin SWE in melt season (Fig. 6c), likely due to decreased sublimation of the intercepted snow on the canopy. In contrast, deforestation of willow biome decreases SWE and length of the snowcover season (Fig. 6c), likely due to decreased snow transport. Different effects of a single vegetation change mechanism (e.g., deforestation in RME) in different biomes highlight the important role of vegetation in influencing snow regimes and particularly, snow transport and sublimation

from the canopy. The first scenario (Fig. 6), which includes all of the potential changes in vegetation cover of each basin, is selected for all the analyses and comparison of the hydrological responses to vegetation, soil, and climate changes.




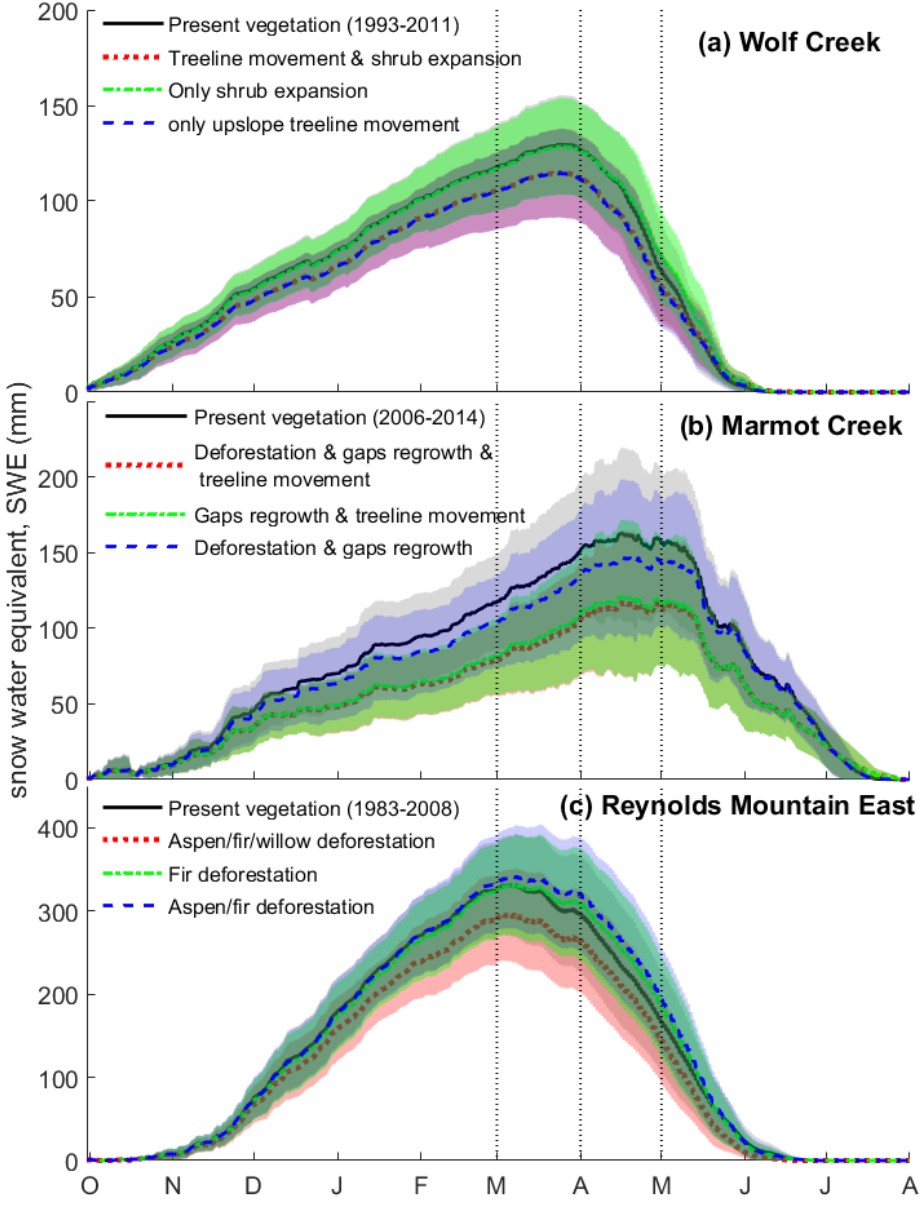

**Figure 6.** Differences in the annual cycle of simulated snow water equivalent (SWE) in the present climate/vegetation and three vegetation change scenarios in the three basins along the North American Cordillera. No climate and soil changes are considered with transient vegetation changes. All three simulated SWE distributions (with hourly time step for the entire record period) under three vegetation change scenarios are significantly ($p$-value $\leq 0.05$) different from the simulated SWE distribution for the present simulations (CVS_) in each basin, based on the Kolmogorov-Smirnov test. The 95 % confidence intervals shown by the shaded areas indicate the interannual variability. Three vertical lines denote the first days of March, April, and May.



### 3.2 Snow characteristics

Simulated snow characteristics including peak SWE, length of the snow season, snow initiation date, mean annual peak SWE, and
timing of snow-free date under current climate and future vegetation and soil in three basins along the NAC were studied under
imposed climate and vegetation changes. Because only three scenarios are applied for future vegetation, only the range of changes
is provided. As it was expected soil changes did not affect snow characteristics and vegetation changes only decreased peak SWE
in MCRB (Table 2). The timing of snowcover season was found to be insensitive to vegetation and soil changes under present
climate (Table 2). Since the model is deterministic, soil modules that do not affect the snow produce identical results (Table 2
columns 2&3 and 4&5). The basin-scale peak SWE is affected by both climate and vegetation changes, and the changes are
statistically significant based on the Mann–Whitney U-test ($p$-values $\leq 0.05$). The difference between distributions of the
simulated present peak SWE and the future distributions obtained from eleven regional climate models ($11\times n$ values) for $n = 18$
years for WCRB, 9 years for MCRB, and 25 years for RME are assessed. Table 3 shows that it decreases from 133 mm under the
current climate to 118 mm (11 % decrease) under a climate change scenario in WCRB. Again, soil modules that do not affect the
snow produce identical results (in Table 3, VS_C = C_VS and S_CV = _CVS). The peak SWE decreases to 107 mm (20 %) when
a transient vegetation change scenario is also considered in combination with a climate change scenario. In the central basin, the
peak SWE declines from a current climate SWE of 183 mm to 141 mm (23 % decrease) under climate perturbation and declines
to 106 mm (42 % decrease) under a combined climate and transient vegetation change. An increase in precipitation in the north
and a large vegetation change in MCRB and its effect on accumulated snow lead to almost an equal peak snowpack in both MCRB
and WCRB. The peak SWE in the southern basin decreases from 368 mm under the current climate to 196 mm (47 % decrease)
under climate perturbation and decreases to 168 mm (54 % decrease) under both climate and vegetation changes. Considering
only vegetation changes under the current climate, the peak SWE decreases more in MCRB (26 %) than in WCRB and RME (11
%). Therefore, under the combined climate and vegetation change studied in this research, the maximum accumulated snowpack
is the most resilient in WCRB and the most sensitive in RME.





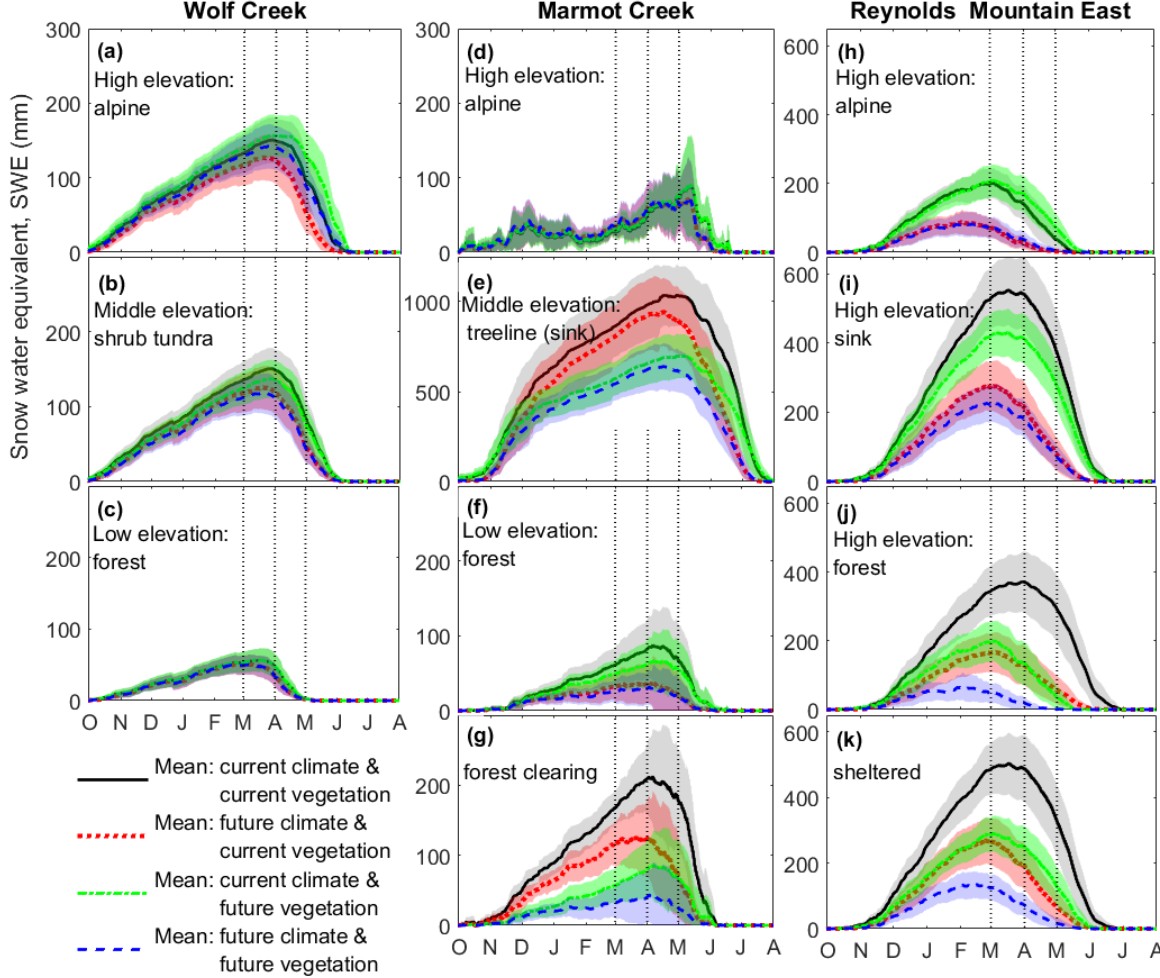

**Figure 7.** Simulated snow accumulation and ablation as snow water equivalent under current climate and vegetation and climate changes in different elevation levels and biomes in Wolf Creek Research Basin, Marmot Creek Research Basin, and Reynolds Mountains East along the North American Cordillera. Reynolds Mountains East has only one elevation band but multiple blowing snow regimes. The 95 % confidence intervals shown by the shaded areas indicate the interannual variability. Ensemble uncertainty is not considered and ensemble mean is instead selected for analyzing the interannual SWE under current and perturbed climatic conditions. Three vertical lines denote the first days of March, April, and May. All three simulated SWE distributions (with hourly time step for the entire record period) under three climate, vegetation, and combined climate and vegetation changes are significantly ($p$-value $\leq 0.05$) different than simulated SWE distribution for the present simulations (CVS_) in each biome, based on the Kolmogorov-Smirnov test.





The monthly variability of SWE with elevation is shown in Fig. 7 (blowing snow regimes in RME). The high, middle, and low elevations each match a vegetation type under present climate. Under future climate, however, each elevation band may include two types of vegetation. For comparing the snow regime changes, each basin was divided into three elevation bands each covering

multiple HRUs. In addition to different elevation bands, the snowpack accumulation and ablation in a forest clearing in MCRB and in a sheltered site in RME are examined. Shaded areas in Fig. 7 show the interannual variability of snowpack under (i) current climate and vegetation, (ii) future climate and current vegetation, (iii) current climate and future vegetation, and (iv) future climate and future vegetation within the three headwater basins across the NAC. Under the vegetation change scenario with an unchanged climate, the peak snow close to a latitude of 61° N becomes deeper, and snowpack ablates slower, and snowcover season becomes

longer (Fig. 7a). This is due to shrub expansion to higher elevations, which reduces snow transport and subsequent sublimation from blowing snow. In contrast, less snow accumulates under future climate with unchanged vegetation and ablation rates are greater. Under a combined climate and vegetation change scenario, the effect of climate change on the alpine snowpack is moderated in the northern basin by the impact of the shrub tundra expansion to higher elevations. At middle elevations, shrubs are expected to be replaced from below by treeline upward movement; therefore, under the vegetation change scenario and the

combined climate and vegetation change scenario, peak snowpack decreases from 156 mm in the current climate to 127 mm (19 % decrease, Fig. 7b). Vegetation change is negligible at low elevations in WCRB. Therefore, the snowpack is not disturbed by vegetation change but is disturbed by climate perturbation in these simulations (Fig. 7c).

In the Canadian Rocky Mountains, MCRB (≈51° N), an upward movement of the treeline causes an increase in the simulated peak snowpack with slower ablation rates at high elevations (Fig. 7d). In contrast to the snowpack enhancement under treeline upward

movement, climate perturbation slightly decreases the peak SWE in the alpine in MCRB. The treeline has an important role in snow redistribution and acts as a blowing snow sink. The effect of treeline rise on the alpine snowpack is greater than that of climate perturbation alone. This is due to the higher sublimation rate of forest in comparison to shrubs and alpine vegetation in the treeline zone. Even though the drifted snow regime in the treeline zone and the alpine snow regime in MCRB are the most resilient biomes to climate perturbation amongst all other biomes in the northern and southern NAC, these biomes are susceptible

to the combined impact of climate perturbation and upward movement of the treeline (Fig. 7e). At low elevations in MCRB, snow accumulation decreases from 87 mm to 39 mm (48 mm) under climate change and conversion of forest to shrub and grass (Fig. 7f) due to potential wildfire and mountain pine beetle (Macias-Fauria and Johnson, 2009). This is because of the shift in the forest role from slowing snowmelt by shading the snow and sheltering the snow from wind to accelerating midwinter snowmelt by removal of the forest canopy (Lundquist et al., 2013). Forest clearings store deep snowpacks under current climate; however, with

regrowth of harvested forest, the peak snow will decrease (Fig. 7g). The impact of climate change is less important than the impact of forest regrowth in the harvested clearings due to increased interception losses. Another impact of forest regrowth is the delay in snow ablation because of the lower net radiation under the canopy relative to clearings with no canopy. In general, the impact of vegetation change is offsetting the impact of climate perturbation on peak snowpack timing. The date of the peak SWE is delayed with only vegetation conversion in MCRB and advanced with only climate change impacts.

In RME (≈43° N), all blowing snow regimes except for the depressions and valley bottom (Fig. 7i) will receive a uniform SWE under vegetation changes as the forest canopy disappears. Despite the small impact of vegetation change in the alpine biome covered with grass and short mountain sages, the impact of climate change on snowpack in this biome is large (Fig. 7h). The most sensitive biome in RME to both climate and vegetation changes is the forest, based on a large decrease in the peak snowpack (Fig. 7j). The interannual variability of SWE, which is expressed as 95 % confidence intervals in Fig. 7, becomes smaller in all of the



biomes within the three basins under climate perturbation because the snowpack becomes shallow under the combined climate and vegetation change and variability of the shallow snowpack becomes smaller. This can occur despite an increased variability of precipitation under the future climate conditions. The interannual variability of SWE does not change in the alpine biomes under a transient vegetation change scenario with an unchanged climate.

Snow regimes are the most resilient to both climate and transient vegetation changes at high elevations in WCRB and MCRB and
low elevations in WCRB, with less than 10 % decrease in the annual peak SWE. In contrast, snow regimes in the forest clearings in MCRB and in the forest and sheltered sites in RME are sensitive to the changes, with 80 % and 68 % decreases, respectively. Under transient vegetation change, the peak SWE drops from 87 mm to 46 mm (47 % decrease) at low elevations in MCRB with the expansion of the grasslands into the forest. Impacts of climate change on snow regimes can be enhanced or dampened by the impact of transient vegetation changes. Shrub tundra expansion into the higher elevations in WCRB can substantially dampen the
impact of climate change on snowpack. Active vegetation growth in the treeline or in forest clearings enhances climate change impacts on the snowpack. Therefore, the impact of afforestation on the snowpack can be as important as the impact of climate change. Details of the differences in snow, sublimation, evapotranspiration, and runoff processes under monthly perturbed climate and transient changes in vegetation and soils are discussed in the next sections and the Mann–Whitney U-test is applied to test whether differences between current climate and future climate/vegetation variables are statistically significant or not. The null
hypothesis is that the distribution mean under current climate is equal to the distribution mean under climate and vegetation change scenarios. The alternative hypothesis is that the climatological means are not equal.

### 3.3 Precipitation Phase

With warmer air temperatures and increased precipitation, the snowfall events become less frequent as the precipitation phase
changes from snowfall to rainfall. As shown in Fig. 8 for three basins, and Fig. 9 for biomes within each basin, the rain to total precipitation ratio, increases in all of the basins under climate and transient vegetation changes. Furthermore, the annual rainfall reaches 238 mm out of 413 mm total annual precipitation, 0.58 in WCRB, 550 mm out of 1027 mm total precipitation, 0.54 in MCRB, and 473 mm out of 866 mm total precipitation, 0.55 in RME. Based on the results for the three basins constituting a north–south transect through the NAC, snow-dominated regions with elevations ranging between 650 m and 2500 m are expected to
become rain-dominated under similar monthly perturbed climatic conditions to these basins.

### 3.4 Snow Transport

Under climate and transient vegetation changes, the annual average snow transport remains unchanged in WCRB, while it declines 14 mm in MCRB and 11 mm in RME (Fig. 8). Snow drifting at high elevations in MCRB declines 11 mm under climate perturbation and increases 23 mm due to shorter fetches as the treeline is moving upslope. Therefore, the impact of climate perturbation
on snow transport in the alpine biome in MCRB is almost completely offset by vegetation change. Under both climate and vegetation changes at middle elevations in MCRB, where the treeline currently exists, snow transport decreases 56 mm under climate change. Snow transport in the valley bottom and blowing snow sink regime in RME, now covered with a willow forest, also




decreases substantially from present-day 79 mm to 37 mm (42 mm decrease, $p$-value $\leq 0.05$) under climate change and deforestation.

**Figure 8.** Mean modelled water fluxes, in three states of liquid, vapor, and snow, under current climate and current vegetation (CVS_), future climate and current vegetation (VS_C), (c) current climate and future vegetation (CS_V), and future climate and future vegetation (S_VC) in Wolf Creek Research Basin, Marmot Creek Research Basin, and Reynolds Mountains East. The statistically significant changes in the climatological mean of the simulated variables with $p$-values less than 0.05 are represented by bold and black values.





**Figure 9.** Mean modelled water fluxes, in three states of liquid, vapor, and snow, on an elevation/ vegetation basis under current climate and current vegetation (CVS_ ), future climate and current vegetation (VS_C), current climate and future vegetation (CS_V), and future climate and future vegetation (S_VC) in Wolf Creek Research Basin, Marmot Creek Research Basin, and Reynolds Mountains East. The statistically significant changes in the climatological mean of the simulated variables with *p*-values less than 0.05 are represented by bold and black values.




### 3.5 Sublimation

The total annual sublimation from all sources including snow intercepted on the canopy, snow surface, and blowing snow was examined under climate and transient vegetation changes (Fig. 8 and 9). Sublimation from snow intercepted on the canopy in

WCRB dominates the total sublimation, which is expected to increase in this basin as the treeline moves upward and shrub tundra expands to higher elevations. In MCRB, total annual sublimation increases 14 mm under vegetation changes but decreases 8 mm under both vegetation and climate changes. The impact of vegetation on sublimation rate in RME is negligible, while climate perturbation decreases sublimation from 31 mm to 10 mm. Transient vegetation change enhances the sublimation with varying rates on different biomes across the NAC. It causes the amount of sublimation to increase moderately in the central and northern

basins. Transient vegetation change does not affect sublimation magnitudes in the southern basin.

Sublimation losses do not only vary from one basin to another but vary among the different elevation bands within each basin. For instance, at high elevations in WCRB, a shrub tundra expansion enhances the sublimation by increasing the snowpack. In contrast, both snowpack and sublimation decrease under climate perturbation. This shows that, in the alpine biome of WCRB, the impact of transient vegetation change on sublimation can be as important as the impact of climate change and a combined climate and

transient change leads to an unchanged sublimation rate. At middle elevations in WCRB covered currently by shrub tundra, a treeline shift into the shrub tundra biome increases sublimation, while the opposite is true under climate perturbation when snowpack and sublimation both decrease. No changes are expected in the sublimation at low elevations in WCRB. Similar to WCRB, the impact of a combined climate and transient vegetation change on sublimation in MCRB varies with elevation. It causes an 8 mm decrease at high elevations as a result of the upward movement of the treeline, a 12 mm increase in the treeline blowing snow

sink regime as shrubs turn to forest, and a 21 mm decrease at low elevations as forest becomes uncovered and snowpack becomes shallower with warming. Different mechanisms are responsible for these changes; total sublimation decreases in the alpine biome with the upward movement of the treeline as sublimation from blowing snow drops with upslope forest expansion. At middle elevations, bushes are replaced by trees and sublimation from intercepted snow on their canopy slightly increases. The combination of topographic gradients and types of vegetation plays an important role in snow redistribution and blowing snow sublimation.

The highest wind-driven redistribution of snow and the highest sublimation occurs on leeward slopes, where there is little or no vegetation cover (Liston et al., 2002). At low elevations in MCRB, sublimation from intercepted snow on the canopy decreases as deforestation occurs. This also occurs in the deforested zone in RME in which sublimation significantly decreases from 104 mm to 8 mm as a result of decreased available snow combined with deforestation under climate perturbation.

### 3.6 Evapotranspiration (ET)

The change in vegetation composition also alters the amount of evapotranspiration (ET). The vegetation composition in all three basins in this study is projected to change under future conditions. The simulations show that, under transient vegetation change, annual ET increases 8 mm as a result of afforestation of the clearings and upward movement of the treeline in MCRB. In contrast, ET decreases 14 mm in WCRB and 18 mm in RME. An increase in ET due to climate perturbation can be offset to some degree by a transient vegetation change in WCRB and RME. ET increases the most in MCRB, from 392 mm to 475 mm (83 mm, $p$-value

$< 0.05$), and the least in WCRB, from 130 mm to 142 mm (12 mm), under both vegetation and climate changes. Under a combined climate and transient vegetation change, statistically significant changes in ET are expected in different elevation bands. The increase in ET varies with elevation within each basin and reaches 23 mm at high elevations and 9 mm at low elevations in WCRB,




61 mm at low elevations and 249 mm in the treeline elevations in MCRB, and 32 mm in the forest and 98 mm in the sheltered site in RME (Fig. 9). This shows a high variability of the annual ET in the three basins along the NAC.

**3.7 Runoff characteristics**

The combined soil and vegetation changes decrease annual runoff volume in WCRB and offset the decreasing effect of climate change on annual runoff volume (Table 4, Fig. 3d). Changes in soil and vegetation decrease annual runoff volume in MCRB (Table 4, Fig. 4e). With both climate and vegetation change effects, annual runoff volume decreases in MCRB, while under combined climate – soil changes it increases (Table 4, Fig. 4e). This shows that a combination of all, vegetation, and soil changes have respectively the largest to lowest effect, and climate change has no effect on annual runoff volume in MCRB. In RME, change in annual runoff is evidenced only under current climate and future vegetation (Table 4, Fig. 5e).

Figure 10 shows the average annual hydrographs for the present and future climates simulations under vegetation and soil changes. With vegetation change, high flows are lower in all three basins, particularly in the WCRB hydrograph (Fig. 10a). Climate change shifts high flows to occur earlier in the WCRB hydrograph (Fig. 10b), and leads to no changes in MCRB (Fig. 10d) and a large advancing for RME (Fig. 10f). Areas under the hydrographs represent the annual runoff volumes for current and future climates under vegetation and soil changes (Fig. 10). Climate and soil changes do not cause significant changes in annual runoff volumes in MCRB and RME, while in WCRB climate change increases and soil change decreases the annual runoff volume (Fig. 3d). Shifts in the timing of snow accumulation and snowmelt seasons have important consequences and can change the timing, rate, and amounts of runoff in mountain basins. A combined effect of climate – vegetation change advanced the snow-free date by 14 days in WCRB, 11 days in MCRB, and 46 days in RME (Table 3) and decreased the length of the snowcover season by 9, 37, and 40 days in WCRB, MCRB, and RME, respectively (Table 3). The combined effect of climate and vegetation changes delayed the snow accumulation initiation date by 15 days in MCRB and 14 days in RME (Table 3). The beginning of the melt season, indicated by the timing of peak SWE, advanced 22 days (April 4 to March 13) in WCRB (Table 3), and 34 days (March 10 to February 4) in RME (Table 3). The shift in the timing of the melt season was reflected in the runoff timing (Fig. 10b & f).





**Figure 10.** Differences in annual hydrograph under present climate, present vegetation, and present soil (CVS_), present climate, future vegetation, present soil (CS_V), present climate, present vegetation, future soil (CV_S), present climate, future vegetation, future soil (C_VS), future climate, present vegetation, present soil (VS_C), future climate, future vegetation, present soil (S_VC), future climate, present vegetation, future soil (V_SC), and future climate, future vegetation, future soil (_CVS) in the three basins. The freshet onset occurs earlier in RME and high runoff advances 47 days under S_VC.



## 4 Discussion

The interaction of vegetation, soil, and climate changes can either trigger large changes in snow and runoff regimes or offset each other's effect. For instance, a statistically insignificant increase in peak SWE in the alpine biome in WCRB under vegetation change can become important as that it can offset the climate change effect when a combination of both changes is considered (Fig. 3a). Furthermore, vegetation change decreases annual runoff in WCRB while climate change counteracts the effect of vegetation change and increases the annual runoff (Fig. 3d). The individual effects of soil and climate change effects on annual runoff in MCRB are statistically insignificant, but when they are combined, they enhance each other's effect, leading to a statistically significant increase in annual runoff volume (Fig. 4e). Therefore, the increasing effect of climate change on annual runoff volume is offset by vegetation change effect in WCRB, and it is enhanced by soil change effect in MCRB while the effect of climate change on annual runoff in RME is not significant, and the vegetation change is the main driver of the runoff changes in this basin. A decreasing effect of vegetation change on annual runoff in MCRB is offset by a combined soil and climate change (_CVS and CVS_ are in the same group in Fig. 4e). This suggests that not only climate change but also transient vegetation and soil changes affect hydrological processes in cold regions and small changes can trigger significant hydrological changes when the changes concur. Therefore, consideration of all vegetation, soil, and climate changes in impact studies is necessary (Pielke, 2005), especially in the basins with near-freezing winter air temperatures such as RME as complex and nonlinear vegetation – atmosphere interactions can dampen or amplify climate change (Bonan, 2008). Under combined climate and transient vegetation changes in WCRB, precipitation and rainfall ratio increase (Fig. 8), peak SWE declines (Table 3), ET and sublimation increases (Fig. 8), and snow season period shortens (Table 3), which result in no change in annual total runoff (Fig. 3d). This implies that the climate change effect on increasing annual runoff in WCRB is offset by the vegetation change effect on decreasing annual runoff and increased precipitation effect is offset by increased sublimation and ET in WCRB (Rasouli et al., 2019). Unlike WCRB, annual runoff volume declines under combined climate and transient vegetation changes (S_VC scenario) in MCRB (Fig. 4e), which is due to significant decreases in sublimation and snow transport and increase in ET (Fig. 8 and 9). The response of simulated annual total runoff to climate and vegetation changes varies. Annual runoff increases from RME in the south to WCRB in the north under only climate and both climate – soil changes, consistent with findings of Nijssen et al. (2001). Annual runoff increases with climate change in WCRB (Fig. 3d) and MCRB (Fig. 4e), and decreases with only vegetation or vegetation – soil changes in all three basins, consistent with Bosch and Hewlett (1982), and with only soil changes in WCRB (Fig. 5e). Despite the snow regime in RME, which is sensitive to both climate and transient vegetation changes, only vegetation change affects annual total runoff. Transient vegetation change moderates the impact of climate change on ET to some extent by decreasing ET in WCRB and RME (Fig. 8). Under a combined climate and transient vegetation change, ET increases in the three basins across the NAC (Fig .8). The response of the peak SWE to climate and transient vegetation changes leads to a complex response of the annual runoff when soil and precipitation phase changes are also considered. Changes in runoff characteristics become statistically significant when combined climate – vegetation – soil changes occur in RME, climate – soil changes occur in MCRB, and soil – vegetation changes occur in WCRB (Fig. 4e and Fig. 5e).

A deep snowpack is deposited at middle elevations in MCRB because of the strong winds, which scour blowing snow from the higher elevations to the treeline (Fang et al., 2013). Under the simulations presented in this paper and ongoing vegetation growth, alpine vegetation and shrubs in the treeline will eventually convert to forest, which can change the snow regime from a present-day blowing snow sink to a future forest with intercepted snow on the canopy. A simulated snow regime change at middle elevations in MCRB leads to a substantial decrease in the maximum accumulated snowpack (Fig. 4c). The peak SWE at low



elevations also declines under future deforestation and climate change in MCRB (Fig. 4b and Fig. 7f). This is because sublimation from blowing snow within the deforested portion of the lower elevations becomes more important than sublimation from intercepted snow on the canopy before deforestation. A higher sublimation rate on the slopes with no vegetation cover was also reported by Liston et al. (2002). The impact of afforestation on snowpack in the forest clearings is stronger than that of climate perturbation. Therefore, an enhanced snowpack decline is expected in forest clearings under climate and vegetation changes (Fig.
4c and Fig. 7g).

Shrub tundra expansion to higher elevations (Myers-Smith and Hik, 2018), community height increase (Bjorkman et al., 2018), and increase of tree growth rates (Innes, 1991) have shifted the blown snow drifts into higher elevations (Fig. 4a; Fig. 6a), which has offset the climate warming effect on decreasing peak SWE in the alpine biome in WCRB (Fig. 3a). A 20–60 % increase in tundra height is expected by the end of the century (Bjorkman et al., 2018), which may change snow redistribution and soil moisture
availability in the higher latitudes. Despite a long snowcover period in higher elevations with shrub tundra expansion, which may slow the growth rate, snow insulates and warms the soil and increases the productivity chance, leading to more expansion of the warmth-demanding vegetation types such as shrub tundra (Lamprecht et al., 2018). The balance of these feedbacks in the future may depend on the changes in air temperature, snow redistribution, and soil moisture and their interactions (Lawrence and Swenson, 2011).

**5 Conclusions**

Hydrological regimes in three headwater basins along the North American Cordillera are vulnerable to climate, vegetation, and soil changes. A physically based semi-distributed hydrological model driven with monthly perturbed climate was reconstructed based on the historical observations and changes in monthly climatology. Changes in monthly climatology were obtained from eleven regional climate models. Climate perturbations, vegetation and soil changes each affect cold regions hydrological
mechanisms. Vegetation changes are large as climate changes and decrease peak SWE at middle elevations, and sublimation amounts. Shrub tundra expansion to higher elevations has shifted the blown snow drifts into higher elevations in the northern basin, which has offset the climate warming effect on decreasing peak SWE in the alpine biome in WCRB. At high elevations, the impact of climate change on peak SWE, snow transport in RME, ET, and annual total runoff is partially offset by the impact of vegetation change.

Simulations in the transect suggest that under both climate perturbation and soil changes annual total runoff is expected to gradually increase from RME in the south to WCRB in the north of the Cordillera. With both vegetation and soil changes annual total runoff will decrease. With all three climate, vegetation, and soil changes simulations suggest that annual runoff will decrease in RME, remain unchanged in MCRB and WCRB. The annual runoff volume decrease under soil change is larger than the annual runoff volume increases under climate change in WCRB. Furthermore, the soil change has a more important role than the vegetation
change in decreasing runoff volume in WCRB. To some extent, the interaction of soil – climate changes moderates the counteracting decreasing effect of soil change and increasing effect of climate change on annual runoff volume. Interaction of soil – climate changes has also a more important role in increasing annual runoff volume than the effect of only climate, only soil change, or interaction of all three soil – vegetation – climate changes in MCRB. Further investigation in other mountainous regions, especially in regions with winter temperatures near freezing is needed to better assess the impact of combined climate, vegetation,





and soil changes. Water resources systems that are vulnerable to warming and land cover changes can be identified using the modelling strategy present here. Future vegetation and soil changes need to be considered in addition to a changing climate to reduce the uncertainties about the future mountain hydrology.

**Acknowledgments.** We thank Danny Marks of USDA, who had a long-term commitment to provide the data for Reynolds Mountain East and the late Rick Janowicz of Yukon Environment, who has provided the Wolf Creek data. The research funding
source is the Natural Sciences and Engineering Research Council of Canada through Discovery Grants, Alexander Graham Bell Canada Graduate Scholarship-Doctoral Program and Postdoctoral fellowship, and Changing Cold Regions Network. We thank Xing Fang for his help on modelling of the Marmot Creek. The discussions and comments of Andrew Ireson and Lawrence Martz on an earlier draft of this paper are appreciated.

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





Tables:

**Table 1.** Comparison of physiography and climatology amongst the three basins across the North American Cordillera. UC denotes Upper Clearing meteorological station in Marmot Creek Research Basin. All three basins are located in transition climate zones based on Köppen (1936) climate classification.

| Characteristics | Wolf Creek | Marmot Creek | Reynolds Mountain East |
|---|---|---|---|
| Latitude | 60°36′N | 50°57′N | 43°11′N |
| Longitude | 134°57′W | 115°09′W | 116°47′W |
| | | | |
| Drainage area [km$^2$] | 179 | 9.4 | 0.38 |
| Elevation range [m] | 660 – 2080 | 1600 – 2825 | 2028 – 2137 |
| Record period | 1993 – 2011 | 2005 – 2014 | 1983 – 2008 |
| | | | |
| Dominant vegetation cover | | | |
| *high elevation* | tundra moss, | rock, grass | grass, sage |
| *middle elevation* | shrub tundra | spruce, fir | fir |
| *low elevation* | spruce | lodgepole pine | aspen, willow |
| | | | |
| Climate zone | Cordillera & | Cordillera & | Cordillera & |
| | sub-Arctic | Prairie & Boreal | Continental & Mediterranean |
| Elevation bands | 3 | 3 | 1 |
| Temperature [°C] | | | |
| *high elevation* | -3.4 | -1.8 | 5.0 |
| *middle elevation* | -2.0 | 1.0 (UC) | - |
| *low elevation* | -1.5 | 2.9 | - |
| Number of Freezing days | | | |
| *high elevation* | 224 | 217 | 120 |
| *middle elevation* | 203 | 166 (UC) | |
| *low elevation* | 179 | 128 | |
| Precipitation [mm] | 380 | 1011 | 858 |
| Wind speed [ms$^{-1}$] | 3.7 | 5.8 | 1.9 |
| Relative humidity [%] | 74 | 69 | 61 |
| | | | |
| Number of sub-basins & HRUs | 5 & 29 | 4 & 36 | 1 & 12 |
| HRU area range [km$^2$] | 0.92 – 25.4 | 0.01 – 1.37 | 0.01 – 0.07 |


**Table 2.** Simulated snow characteristics under current climate and current vegetation and future vegetation and soil in three basins along the North American Cordillera. Because only three scenarios are applied for future vegetation, only the range of changes is provided. Underlined values denote significant changes with *p*-values less than 0.1. Changes relative to current climate/ vegetation are given in parentheses.

| Variable | Current Climate Current Vegetation | Current Climate ΔSoil | Current Climate ΔVegetation | Current Climate ΔSoil&Vegetation |
|---|---|---|---|---|
| **(1) Wolf Creek Research Basin (WCRB)** | | | | |
| Peak SWE [mm] | 133 | 133 | 118-133(-11 to 0) | 118-133(-11 to 0) |
| Initiation [date] | 5 | 5 | 5(0) | 5(0) |
| Peak SWE [date] | 186 | 186 | 182-185(-4 to -1) | 182-185(-4 to -1) |
| Snow-free [date] | 250 | 250 | 250-252(0 to 2) | 250-252(0 to 2) |
| Season length [day] | 224 | 224 | 224-226(0 to 2) | 224-226(0 to 2) |
| **(2) Marmot Creek Research Basin (MCRB)** | | | | |
| Peak SWE [mm] | 183 | 183 | 136-168(-26 to -8) | 136-168(-26 to -8) |
| Initiation [date] | 9 | 9 | 9(0) | 9(0) |
| Peak SWE [date] | 210 | 210 | 211(1) | 211(1) |
| Snow-free [date] | 294 | 294 | 294-296(0 to 2) | 294-296(0 to 2) |
| Season length [day] | 283 | 283 | 283-284(0 to 1) | 283-284(0 to 1) |
| **(3) Reynolds Mountain East (RME)** | | | | |
| Peak SWE [mm] | 368 | 368 | 326-375(-11 to 2) | 326-375(-11 to 2) |
| Initiation [date] | 35 | 35 | 35(0) | 35(0) |
| Peak SWE [date] | 161 | 161 | 162-168(1 to 7) | 162-168(1 to 7) |
| Snow-free [date] | 246 | 246 | 247(1) | 247(1) |
| Season length [day] | 211 | 211 | 212-213(1 to 2) | 212-213(1 to 2) |

| 1st of | Oct | Nov | Dec | Jan | Feb | Mar | Apr | May | Jun | Jul | Aug | Sep |
|---|---|---|---|---|---|---|---|---|---|---|---|---|
| Day of Water Year | 1 | 32 | 62 | 93 | 124 | 152 | 183 | 213 | 244 | 274 | 305 | 336 |





**Table 3.** Simulated snow characteristics under current and monthly perturbed climate and future vegetation in three basins along the North American Cordillera. Bold and underlined values denote significant changes with *p*-values less than 0.05 and 0.1, respectively. Changes, which are relative to current climate/ vegetation, are given in parentheses. Dates are given in Julian water year.

| Variable | Current Climate Current Vegetation | ΔClimate Current Vegetation | | | ΔClimate+ ΔSoil | | | ΔClimate+ ΔVegetation | | | ΔClimate+ ΔSoil &ΔVegetation | | |
|---|---|---|---|---|---|---|---|---|---|---|---|---|---|
| | | 5 % | mean | 95 % | 5 % | mean | 95 % | 5 % | mean | 95 % | 5 % | mean | 95 % |
| **(1) Wolf Creek Research Basin (WCRB)** | | | | | | | | | | | | | |
| Peak SWE [mm] | 133 | 73 | 118 (-11) | 153 | 73 | 118 (-11) | 153 | 64 | **107 (-20)** | 142 | 64 | **107 (-20)** | 142 |
| Initiation [date] | 5 | 0 | 7 (2) | 47 | 0 | 7 (2) | 47 | 0 | 7 (2) | 45 | 0 | 7 (2) | 45 |
| Peak SWE [date] | 186 | 143 | **164 (-22)** | 178 | 143 | **164 (-22)** | 178 | 148 | **164 (-22)** | 170 | 148 | **164 (-22)** | 170 |
| Snow-free [date] | 250 | 213 | **235 (-15)** | 248 | 213 | **235 (-15)** | 248 | 216 | **236 (-14)** | 249 | 216 | **236 (-14)** | 249 |
| Season length [day] | 224 | 160 | **208 (-16)** | 242 | 160 | **208 (-16)** | 242 | 164 | **215 (-9)** | 251 | 164 | **215 (-9)** | 251 |
| **(2) Marmot Creek Research Basin (MCRB)** | | | | | | | | | | | | | |
| Peak SWE [mm] | 183 | 102 | **141 (-23)** | 170 | 102 | **141 (-23)** | 170 | 74 | **106 (-42)** | 130 | 74 | **106 (-42)** | 130 |
| Initiation [date] | 9 | 4 | **24 (15)** | 62 | 4 | **24 (15)** | 62 | 4 | **24 (15)** | 63 | 4 | **24 (15)** | 63 |
| Peak SWE [date] | 210 | 175 | 200 (-10) | 216 | 175 | 200 (-10) | 216 | 177 | 205 (-5) | 223 | 177 | 205 (-5) | 223 |
| Snow-free [date] | 294 | 257 | **281 (-13)** | 295 | 257 | **281 (-13)** | 295 | 257 | **283 (-11)** | 299 | 257 | **283 (-11)** | 299 |
| Season length [day] | 283 | 204 | **248 (-35)** | 277 | 204 | **248 (-35)** | 277 | 200 | **246 (-37)** | 276 | 200 | **246 (-37)** | 276 |
| **(3) Reynolds Mountain East (RME)** | | | | | | | | | | | | | |
| Peak SWE [mm] | 368 | 105 | **196 (-47)** | 277 | 105 | **196 (-47)** | 277 | 91 | **168 (-54)** | 237 | 91 | **168 (-54)** | 237 |
| Initiation [date] | 35 | 20 | **50 (15)** | 85 | 20 | **50 (15)** | 85 | 19 | **49 (14)** | 83 | 19 | **49 (14)** | 83 |
| Peak SWE [date] | 161 | 102 | **129 (-33)** | 148 | 102 | **129 (-33)** | 148 | 96 | **127 (-34)** | 149 | 96 | **127 (-34)** | 149 |
| Snow-free [date] | 246 | 184 | **213 (-33)** | 232 | 184 | **213 (-33)** | 232 | 195 | **220 (-26)** | 236 | 195 | **220 (-26)** | 236 |
| Season length [day] | 211 | 113 | **161 (-50)** | 197 | 113 | **161 (-50)** | 197 | 129 | **171 (-40)** | 200 | 129 | **171 (-40)** | 200 |

| 1st of | Oct | Nov | Dec | Jan | Feb | Mar | Apr | May | Jun | Jul | Aug | Sep |
|---|---|---|---|---|---|---|---|---|---|---|---|---|
| Day of Water Year | 1 | 32 | 62 | 93 | 124 | 152 | 183 | 213 | 244 | 274 | 305 | 336 |





723
724
725
726
727

**Table 4.** Simulated runoff characteristics including annual volume under current and monthly perturbed climates and future vegetation in basins along the North American Cordillera. Bold and underlined values denote significant changes with $p$-values less than 0.05 and 0.1, respectively, based on the Mann-Whitney U-test. Simulated distributions with $n = 18$ years for WCRB, 9 years for MCRB, and 25 years for RME over the control (Base) period for each hydrological variable are compared with the simulated future distributions obtained from eleven regional climate models ($11 \times n$ values). Percentage change, which are relative to the current climate/vegetation, are given in parentheses.

| | Wolf Creek Research Basin (WCRB) | Marmot Creek Research Basin (MCRB) | Reynolds Mountain East (RME) |
|---|---|---|---|
| Simulation years | 18 | 9 | 25 |
| CVS_ | 246 | 402 | 371 |
| CS_V | 228-262 (-7 to +7) | 336-373(-16 to -7) | 340-379(-8 to +2) |
| CV_S | 210 (-15) | 335 (-17) | 331 (-11) |
| C_VS | **173 (-30)** | 411 (2) | 365 (-2) |
| VS_C | 286 (16) | 426 (6) | 375 (1) |
| S_VC | 265 (8) | 359 (-11) | 351 (-5) |
| V_SC | 250 (2) | 414 (3) | 342 (-8) |
| _CVS | 282 (15) | **492 (22)** | 368 (-1) |

728