# Peer review of "Are the effects of vegetation and soil changes as important as climate change impacts on hydrological processes?"

_Hydrology and Earth System Sciences, 2019_

## Referee Comment (RC1) · Anonymous Referee #1 · 1 Jul 2019

This paper presents a study of changes to hydrological processes under shifting climate, soils and vegetation. The paper is very interesting and the manuscript presents a novel, challenging and important study. However, the current presentation of the work makes it very difficult to really understand what is being said and follow the (complex) findings of the work. I have made some suggestions for the authors to consider on how to organize the presentation that may assist the reader to better follow, understand, and derive a meaningful message from this work.

The Introduction is not well organized and I suggest rewriting and reorganizing it to develop the argument on why this study is necessary, and why it is important. For

example, the very first line of the Introduction discusses the French Alps. Is the paper about the Alps? I don't think it is. The rest of this paragraph is devoted to a somewhat rambling discussion on vegetation and soil changes that may shift under climate change. Methods (i.e. descriptions of scenarios) are scattered throughout this paragraph, and also through various sections of the Introduction. These should be moved to the Methods section.

The last sentence in the first paragraph (line 49) of the paper "Vegetation changes can alter soil properties." is out of place and probably belongs in the first sentence of a paragraph on this topic.

Second paragraph (line 50), suggest moving the second sentence as the opening sentence of the paragraph. This paragraph also jumps around a lot, and needs reorganizing.

Third paragraph of the paper (line 61) is disjointed. Can you incorporate these ideas into the paragraphs above?

Fourth paragraph, this part of the paper you are trying to make an argument for why you use the delta method on your historical observations, but it isn't clear. You talk about noise (line 70). What do you mean by this? This whole argument needs to be clearer and emerge from an explanation of what you have done. Right now it seems out of place.

Line 80, fifth paragraph. This is a strange paragraph. Of course, there have been lots of studies on climate change on hydrology and on mountain hydrology. Is it necessary? Are you planning on summing up all of the results in the field? I think you want to make the argument that lots of these studies have been done but most are missing the vegetation / soil / land surface change component. Perhaps rethink this approach and place your argument in that context.

Line 90-115. These are almost methods to me. I don't think this text belongs in the

Introduction section.

Because you gloss over some things, the paper is difficult to follow. For example, you don't really describe the model so all the figures are tough to follow. What is "drift in"? You show this in the figures but it isn't explained (or I missed it). I think this is because you have not really described the model in the paper. I think you need to describe some bare minimum so the reader can follow your results.

Figure 1. Should this be Figure 1? I think this figure belongs in methods, not in the Intro.

Line 132. You barely describe the data input used to parameterize the model. I think you need some more detail here on what you did. Or, perhaps these sections need reorganizing so it is more clear what was done.

Line 146. What do you mean by "allow differentiation"?

Line 150. This sentence starting with These CHRM models… is unnecessary and can be removed.

Line 154. HRU have different sizes. This is an awkward sentence. Is it necessary?

Line 156-164. This is a really important section to be clear, and it is not. I don't understand what you did, how you modified the HRUs for each scenario, and that is kind of the main point of the paper. I would suggest rewriting this paragraph.

Line 170. Global climate models

Line 176. You start using your notation before you have explained it.

Section 2.1 and end of Section 2.4. I would strongly suggest you rethink how you are notating the paper's findings. You use so many acronyms that make it really difficult to read the paper. Could you think about using actual words for these acronyms? Perhaps Wolf, Marmot and Reynolds for the study sites? The same comment applies for your scenarios. You also use groups in two ways (I think). So, now you have a, b, c in the

figures. On top of the CVS_ /CSV, etc notation. It is a complex study to follow to begin with, so you need to make it easier for the reader (and the reviewers!) to follow.

Line 208 each other's effects. You use this personification a few times in the paper and it is odd.

Figure 2, I think you have the space here to just label the three sites in the Figure.

Discussion, first paragraph, is too long. Break it up to make it easier to digest/understand/follow.

Line 491, "...such as RME where changes are complex and nonlinear...

Table 3, needs to be organized. Can you add in lines/cells?

General

I don't think you make the argument well for changing soils under climate change. This is an important argument to make strongly to support the need for the level of detail in the results. Suggest looking carefully at this.

The initial paragraph of the paper, and some sections of the paper (Discussion, paragraph 1, line 478-509!) have extremely long paragraphs. These should be shorted and broken up. Try to think about the main point you want to make in a paragraph, and let that lead your writing.

Would you consider using a first person voice for this text? I wonder if it might help with some of the awkwardness of the text.

This paper is really long. Suggest to think about all the figures, tables, and the results section (which is ~15 pages) and see where you could reduce the text? Think about each sentence you use and ask yourself if the reader needs to know this information? Why is it important? Can I be more clear? What could be moved to a Supplement section?
I am happy to review this paper again and get to some of the details once the paper has been reorganized and these comments addressed.

---

## Referee Comment (RC2) · Anonymous Referee #2 · 9 Jul 2019

The paper addresses a very relevant topic with high scientific and applied implications. Used methodology is robust and results of high interest. However, I agree wit reviewer 1 that the paper is very difficult to be read because of excessive information on the one side, and because current structure is currently unclear. I think is necessary to select more the information and to facilitate to readers the lecture. Once this will be achieve, it will be a great contribution for the Journal and most important to the field of mountain hydrology.

Specific comments.

-The abstract is not very informative now, it does not inform about the sign and magnitude of predicted changes.

-Line 9. I would not say "seldom studied" Impact of climate-vegetation changes on hydrology have been widely studied in many areas; the most novel of the study is to focus on snow dominated basins. -Line 14. Not sure if "but" is appropriate here. I would say ...SWE "and" increased evapotranspiration. -Line 16. It is not stated before that soils have been also perturbed.

- Introduction needs better organized. The literature review are mixed with the objectives. I would detail the objectives at the end of the section. - Paragraph in lines 45-50 needs to be better organized. -Lines 57-60 are highly repeated with previous paragraph. -Lines 65-80 can be moved to methodology. - Can you incorporate in Figure 1 the applied changes to soils? - Line 157- "changed" instead of "changes". - I would convert lines 185-200 into a table. -Section 2.3. Did you perturbed T and P, or all the variables?

- Figure 1 is absolutely necessary to understand methodology, may be you can use a similar template to provide a fast view of the most important hydrological changes at each site and under different environmental changes. - Many parts of Results are in reality discussion. I would separate better the contents or I would create a results and discussion section.

-Line 219. Turkey's test should be presented in Methods section. -Line 298. It is interesting to see snow cover insensitive to vegetation when many studies point out the opposite.

- I do not see the point of a section 3.2 about snow characteristics when 3.1 also presents changes on snow.

- Are the hypsometry of the three catchments similar or different? how this may affect the results?. - MCRB is the only with predicted deforestation; is this the reason why snow is the most resilient to CC? - Are normal the very low values of sublimation in

WC and RC?

I hope my comments will result useful when preparing the revised manuscript.

———————————————

---

## Author Comment (AC1) · 14 Oct 2019

Response to comments on HESS-2019-214-RC1

Referee #1

This paper presents a study of changes to hydrological processes under shifting climate, soils and vegetation. The paper is very interesting and the manuscript presents a novel, challenging and important study. However, the current presentation of the work makes it very difficult to really understand what is being said and follow the (complex) findings of the work. I have made some suggestions for the authors to consider on how

to organize the presentation that may assist the reader to better follow, understand, and derive a meaningful message from this work.

\*\*\*\*\*AC: The comments and suggestions of Reviewer #1 were very helpful in improving this manuscript. We have rewritten the manuscript to make the presentation of a complex study easier to follow. We have reduced the number of acronyms, using the basin names instead of letter codes and use a consistent letter code to refer to the eight cases that cover the present and future climates, vegetation, and soils.

-The Introduction is not well organized and I suggest rewriting and reorganizing it to develop the argument on why this study is necessary, and why it is important. For example, the very first line of the Introduction discusses the French Alps. Is the paper about the Alps? I don't think it is. The rest of this paragraph is devoted to a somewhat rambling discussion on vegetation and soil changes that may shift under climate change. Methods (i.e. descriptions of scenarios) are scattered throughout this paragraph, and also through various sections of the Introduction. These should be moved to the Methods section.

\*\*\*\*\*AC: We have rewritten the introduction in response to the reviewer's suggestions. The order of presentation has been changed so the three aspects of this study: vegetation, soil and climate change, are dealt with sequentially. The basins are referred to by name as are the biomes in each. The manner in which the eight "treatment" cases are referred to have been made consistent throughout the paper, and we have tried to increase the clarity by more frequent reference to the case designators.

We have gathered various statements referring to the methodology from the manuscript and captions into one coherent Methods section on the recommendation of the reviewer.

Figures and Tables have been revised to be consistent with terms and groupings and order of presentation. Figure legends now use the same designators in the same order replacing several different orders and formats.

-The last sentence in the first paragraph (line 49) of the paper "Vegetation changes can alter soil properties." is out of place and probably belongs in the first sentence of a paragraph on this topic.

*****AC: Addressed in the rewrite. This is now the topic sentence for the third paragraph in the Introduction.

-Second paragraph (line 50), suggest moving the second sentence as the opening sentence of the paragraph. This paragraph also jumps around a lot, and needs reorganizing.

*****AC: Addressed in the rewrite.

-Third paragraph of the paper (line 61) is disjointed. Can you incorporate these ideas into the paragraphs above?

*****AC: Addressed in the rewrite.

-Fourth paragraph, this part of the paper you are trying to make an argument for why you use the delta method on your historical observations, but it isn't clear. You talk about noise (line 70). What do you mean by this? This whole argument needs to be clearer and emerge from an explanation of what you have done. Right now it seems out of place.

*****AC: Addressed in the rewrite.

-Line 80, fifth paragraph. This is a strange paragraph. Of course, there have been lots of studies on climate change on hydrology and on mountain hydrology. Is it necessary? Are you planning on summing up all of the results in the field? I think you want to make the argument that lots of these studies have been done but most are missing the vegetation / soil / land surface change component. Perhaps rethink this approach and place your argument in that context.

*****AC: Addressed in the rewrite.

-Line 90-115. These are almost methods to me. I don't think this text belongs in the Introduction section.

*****AC: Moved to the new Methods section.

-Because you gloss over some things, the paper is difficult to follow. For example, you don't really describe the model so all the figures are tough to follow. What is "drift in"? You show this in the figures but it isn't explained (or I missed it). I think this is because you have not really described the model in the paper. I think you need to describe some bare minimum so the reader can follow your results.

*****AC: Addressed in the rewrite. We have made a large number of changes in the use of terminology so that it is consistent and does not include unnecessary synonyms that could confuse the reader. This should make the text, which we appreciate is complex, more clear.

-Figure 1. Should this be Figure 1? I think this figure belongs in methods, not in the Intro.

*****AC: The reviewer is correct. This Figure has been placed in the Methods section.

-Line 132. You barely describe the data input used to parameterize the model. I think you need some more detail here on what you did. Or, perhaps these sections need reorganizing so it is more clear what was done.

*****AC: Addressed in the rewrite. We have rewritten this section and added more details and references to both the data and modelling strategy that have been published separately.

-Line 146. What do you mean by "allow differentiation"?

*****AC: Addressed in the rewrite. See lines 145 – 155 in the revised manuscript.

-Line 150. This sentence starting with These CHRM models: : : is unnecessary and can be removed.

*****AC: Removed.

-Line 154. HRU have different sizes. This is an awkward sentence. Is it necessary?

*****AC: Removed.

-Line 156-164. This is a really important section to be clear, and it is not. I don't understand what you did, how you modified the HRUs for each scenario, and that is kind of the main point of the paper. I would suggest rewriting this paragraph.

*****AC: This text was rewritten so the changes that were made to the HRU's were made clear. The entire paragraph was rewritten. See lines 163 – 167 in the revised manuscript.

-Line 170. Global climate models

*****AC: "general circulation models" was replaced with "global climate models"

-Line 176. You start using your notation before you have explained it.

*****AC: Addressed in the revision. We have used a simplified notation that only covers what was changed in a specific case. We have also added text that specifically states the cases and columns have been added to tables to include the notation, and the same notation is used in the legends of figures. Also, the order has been made consistent throughout so the reader can follow it more easily.

-Section 2.1 and end of Section 2.4. I would strongly suggest you rethink how you are notating the paper's findings. You use so many acronyms that make it really difficult to read the paper. Could you think about using actual words for these acronyms? Perhaps Wolf, Marmot and Reynolds for the study sites? The same comment applies for your scenarios. You also use groups in two ways (I think). So, now you have a, b, c in the figures. On top of the CVS_ /CSV, etc notation. It is a complex study to follow to begin with, so you need to make it easier for the reader (and the reviewers!) to follow.

*****AC: We now use Wolf Creek, Marmot Creek, and Reynolds Mountain in the text

instead of acronyms.

We have used a simplified notation that only covers what was changed in a specific case using $\Delta C$ for changed climate, $\Delta V$ for changed vegetation, $\Delta S$ for changed soil and the other permutations to cover the eight cases. We now refer to these as cases and not scenarios to avoid confusing the reader with SRES scenarios and RCPs.

All the text in both sections improved and the indices in the text were retained parenthetically for easier reading of the text. All of the related figures were revised to address these comments.

-Line 208 each other's effects. You use this personification a few times in the paper and it is odd.

*****AC: The text now avoids personification.

-Figure 2, I think you have the space here to just label the three sites in the Figure.

*****AC: We now use Wolf Creek, Marmot Creek, and Reynolds Mountain in the Figures instead of acronyms.

-Discussion, first paragraph, is too long. Break it up to make it easier to digest/understand/follow.

*****AC: Addressed in the revision. The text was broken into logical sections.

-Line 491, ": : :such as RME where changes are complex and nonlinear: : :

*****AC: Addressed in the revision.

-Table 3, needs to be organized. Can you add in lines/cells?

*****AC: Table 3 was revised to improve the organization. Space was added between the basins

General

-I don't think you make the argument well for changing soils under climate change. This is an important argument to make strongly to support the need for the level of detail in the results. Suggest looking carefully at this.

*****AC: Addressed in the revision. We agree that this is an important component of the story and tried to make the explanation and the uncertainties in this clear. See lines 159 – 167 in the revised manuscript.

-The initial paragraph of the paper, and some sections of the paper (Discussion, paragraph 1, line 478-509!) have extremely long paragraphs. These should be shorted and broken up. Try to think about the main point you want to make in a paragraph, and let that lead your writing.

*****AC: Addressed in the revision. Paragraphs length has been considered carefully in the revision and where possible different topics or subtopics separated to reduce paragraph length.

-Would you consider using a first person voice for this text? I wonder if it might help with some of the awkwardness of the text.

*****AC: We understand that there is nothing wrong with using first-person voice. All three co-authors, however, prefer to use passive voice style for scientific writing.

-This paper is really long. Suggest to think about all the figures, tables, and the results section (which is ∼15 pages) and see where you could reduce the text? Think about each sentence you use and ask yourself if the reader needs to know this information? Why is it important? Can I be more clear? What could be moved to a Supplement section?

*****AC: This was addressed in the revision. We removed one of the figures and the text related to it. In other places we have simplified the presentation. We have added information that a reader should have to be able to better understand the presentation and the study. We have also referenced our 2019 paper, upon which this paper builds,

so a reader could seek more details that are provided therein.

-I am happy to review this paper again and get to some of the details once the paper has been reorganized and these comments addressed.

*****AC: Thank you for your insightful and detailed comments which were accepted and have been most useful in improving the manuscript.

---

## Author Comment (AC2) · 14 Oct 2019

Response to comments on HESS-2019-214-RC2

Referee #2

The paper addresses a very relevant topic with high scientific and applied implications. Used methodology is robust and results of high interest. However, I agree wit reviewer 1 that the paper is very difficult to be read because of excessive information on the one side, and because current structure is currently unclear. I think is necessary to select more the information and to facilitate to readers the lecture. Once this will be achieve,

[Figure]

**HESSD**

it will be a great contribution for the Journal and most important to the field of mountain hydrology.

*****AC: We have rewritten the manuscript to make the presentation of a complex study easier to follow. The structure of the manuscript has been simplified and the terminology changed to a simpler and more consistent form. We have modified most of the figures to improve clarity and consistency of the presentation.

Specific comments.

-The abstract is not very informative now, it does not inform about the sign and magnitude of predicted changes.

*****AC: Addressed in the revision. Extraneous material has been removed and the focus placed on describing the simulated effects of changes in climate, vegetation, and soils.

-Line 9. I would not say "seldom studied" Impact of climate-vegetation changes on hydrology have been widely studied in many areas; the most novel of the study is to focus on snow dominated basins.

*****AC: Addressed in the revision. We have emphasized that the novel aspect of this study is snow dominated mountain basins. See line 9 in the revised manuscript.

-Line 14. Not sure if "but" is appropriate here. I would say ...SWE "and" increased evapotranspiration.

*****AC: The word "but" was changed to "and".

-Line 16. It is not stated before that soils have been also perturbed.

*****AC: Addressed in the revision. Text regarding soils in the Introduction has been rewritten and made clear.

- Introduction needs better organized. The literature review are mixed with the objectives. I would detail the objectives at the end of the section.

*****AC: Addressed in the revision.

- Paragraph in lines 45-50 needs to be better organized.

*****AC: Addressed in the revision.

-Lines 57-60 are highly repeated with previous paragraph.

*****AC: Addressed in the revision.

-Lines 65-80 can be moved to methodology. - Can you incorporate in Figure 1 the applied changes to soils?

*****AC: Addressed in the revision. The text was moved to the Methods section, and the changes applied to the soils have been added to Figure 2 [previously Figure 1].

- Line 157- "changed" instead of "changes".

*****AC: Corrected.

- I would convert lines 185-200 into a table. -Section 2.3. Did you perturbed T and P, or all the variables?

*****AC: Thanks for this excellent suggestion. We changed these lines into the new Table 2.

We did perturb P and T and kept relative humidity constant to allow vapour pressure to change with warming. Text was added to the method to make the reader aware of this fact. Also, we note in the text that no change were assumed in the rest of the parameters such as wind as the RCM outs for these variables were highly uncertain for the present climate.

- Figure 1 is absolutely necessary to understand methodology, may be you can use a similar template to provide a fast view of the most important hydrological changes at each site and under different environmental changes.

*****AC: In Figure 2 [formerly Figure 1] we added the modelled change in the snow water equivalent (SWE) under climate change scenario into Figure 2. Because we have eight scenarios, including the present climate-vegetation-soil, and six parameters for each scenario (Peak SWE, timings, annual runoff, etc.), it is not easy to show the results on a figure similar to Figure 1.

- Many parts of Results are in reality discussion. I would separate better the contents or I would create a results and discussion section.

*****AC: Addressed in the revision. Discussion text has been moved from the Results to the Discussion.

-Line 219. Turkey's test should be presented in Methods section.

*****AC: Addressed in the revision. A new paragraph was added to the Methods to describe the test. See lines 195 - 204 in the revised manuscript.

-Line 298. It is interesting to see snow cover insensitive to vegetation when many studies point out the opposite.

*****AC: We have emphasized this more strongly in the revision.

- I do not see the point of a section 3.2 about snow characteristics when 3.1 also presents changes on snow.

*****AC: We are of the opinion that these two sections cover very different aspects and chose not to combine them so as to bring out important results.

- Are the hypsometry of the three catchments similar or different? how this may affect the results?.

*****AC: We have shown range of elevation and other physiographic characteristics of the basins in Figure 2 and we discussed the similarities and differences between the basins, which are provided in section 2.1 Study areas and data sources. We acknowledge that there might be some uncertainties in the results due to the different characteristics of the basins, but they are the classic basins for their regions and so we have to use what nature has provided and what research organisations have instrumented.

We have work in progress on the impact of hypsometry that will further address the reviewer's point.

- MCRB is the only with predicted deforestation; is this the reason why snow is the most resilient to CC? - Are normal the very low values of sublimation in WC and RC?

*****AC: We also expect deforestation in Reynolds Mountain as shown in Figure 2 (lower panel) where sage replaces different types of trees.

- Are normal the very low values of sublimation in WC and RC?

*****AC: We have not modified the manuscript in response to this comment, but higher elevations in Marmot Creek are very cold and snowcover period is longer there and a moderate warming does not affect the snow at these elevations (Rasouli et al., 2019a), but it does affect snow at low elevations similar to the deforestation effect.

Total annual sublimation to some extent depends on the annual snowfall and peak SWE. We have less snowpack in Wolf Creek, so we expect lower sublimation in Wolf Creek over an annual cycle and large sublimation in Marmot Creek as it has the highest snow accumulation among the three sites and is subject to Chinook winds.

-I hope my comments will result useful when preparing the revised manuscript.

*****AC: We appreciate your very helpful comments. Addressing your comments and suggestion has helped us prepare a much impr
* * *